# Photosynthetic Carbon Uptake Correlates with Cell Protein Content during Lipid Accumulation in the Microalga *Chlorella vulgaris* NIES 227

Paul Chambonniere *, Adriana Ramírez-Romero, Alexandra Dimitriades-Lemaire, Jean-François Sassi and Florian Delrue

MicroAlgae Processes Platform—CEA, CEA Tech Région Sud, 13108 Saint Paul lez Durance, France
* Correspondence: paul.chambonniere@univ-amu.fr

**Abstract:** Large-scale microalgae cultivation for biofuel production is currently limited by the possibility of maintaining high microalgae yield and high lipid content, concomitantly. In this study, the physiological changes of *Chlorella vulgaris* NIES 227 during lipid accumulation under nutrient limitation was monitored in parallel with the photosynthetic capacity of the microalgae to fix carbon from the proxy of oxygen productivity. In the exponential growth phase, as the biomass composition did not vary significantly (approx. $53.6 \pm 7.8\%$ protein, $6.64 \pm 3.73\%$ total lipids, and $26.0 \pm 9.2\%$ total carbohydrates of the total biomass dry-weight), the growth capacity of the microalgae was preserved (with net $O_2$ productivity remaining above $(4.44 \pm 0.93) \times 10^{-7}$ g $O_2 \cdot \mu$mol PAR$^{-1}$). Under nutrient limitation, protein content decreased (minimum of approx. $18.6 \pm 6.0\%$), and lipid content increased (lipid content up to $56.0 \pm 0.8\%$). The physiological change of the microalgae was associated with a loss of photosynthetic activity, down to a minimum $(1.27 \pm 0.26) \times 10^{-7}$ g $O_2 \cdot \mu$mol PAR$^{-1}$. The decrease in photosynthetic $O_2$ productivity was evidenced to correlate to the cell internal-protein content ($R^2 = 0.632$, $p = 2.04 \times 10^{-6}$, N = 25). This approach could serve to develop productivity models, with the aim of optimizing industrial processes.

**Keywords:** microalgae; *Chlorella vulgaris*; photosynthesis; productivity; biofuel; lipid

## 1. Introduction

Owing to high growth speed and high lipid accumulation capacity, microalgae have recently been heralded as the only feedstock for biodiesel with the capacity to completely displace petroleum-derived liquid fuels [1]. However, after over 15 years of research, biofuel from microalgae have not reached the global market. As of 2021, advanced biofuels represent only 0.3 out of 4.3 EJ of energy produced in the form of liquid biofuel in 2020 [2], of which microalgae are a non-existent to negligible portion (for instance, no data was reported for aquatic biomass use for energy generation by the Bioenergy Europe Statistical Report on Biomass Supply [3]).

Several shortcomings were identified, explaining the divergence between initial expectations and the current state of biofuel production from microalgae. Critically, it has been challenging to convert biomass with a high lipid content into biofuels. Lam et al. (2012) [4] listed various energy efficiency ratios for microalgae-derived biofuel production based on life-cycle analyses, several of which were below 1, with values as low as 0.07 being reported [5]. Microalgae biomass production also demands significant amounts of nutrients [4] and water [6], increasing production costs and compromising the sustainability of the technology. The use of waste streams to reclaim polluting nutrients and grow algal biomass [7,8] could enable the reduction of both the cost and environmental impact of microalgae-derived biofuel [9,10]. Yet another challenge is the discrepancy between early statements on the achievable biomass and lipid productivity, and currently reported results. While Chisti (2007) [1] evaluated a potential for lipid productivity from microalgae in the

range 58.7–137 $m^3 \cdot ha^{-1} \cdot yr^{-1}$ (or a lipid yield of 14.4–33.6 $g \cdot m^{-2} \cdot d^{-1}$), field data collected in pilot-scale photobioreactors operated outdoors demonstrated significantly lower yields. For instance, Xia et al. (2014) [11] found a maximum lipid productivity of 22.8 $m^3 \cdot ha^{-1} \cdot yr^{-1}$ (projecting a yearly production of 10.6 $m^3 \cdot ha^{-1} \cdot yr^{-1}$ if including winter months), and Wen et al. (2016) [12] reported lipid productivities in the range of 2.0–2.9 $g \cdot m^{-2} \cdot d^{-1}$. In addition to the impact of meteorological conditions as predicted by Moody et al. (2014) [13], there exists a dichotomy between algal biomass productivity and lipid accumulation, since the accumulation of lipid by microalgae is fueled by an excess of energy supply at the level of the Calvin cycle, due to an external stress disrupting the cell anabolic activity [14]. Such stresses favoring lipid accumulation in microalgae were listed by Morales et al. (2021) [15], evidencing that the most common stress used by researchers to induce lipid accumulation is nitrogen starvation. Biomass growth is hindered by the stress applied, and significant loss in biomass productivity is expected over the course of lipid production. In order to evaluate the potential for biofuel from microalgae to mature into an applicable technology, it is necessary to characterize microalgae productivity throughout the growth and starvation phases, as the microalgae physiological status evolves.

Based on recent reviews of microalgae productivity models [16,17], few aimed at including the impact of cell physiological changes on biomass productivity, and none of them presented dedicated experiments to explicitly measure the photosynthetic productivity of the organism studied for a given physiological state. There is therefore a need for data linking the instant microalgae productivity with its physiological status.

This study therefore aimed at studying and quantifying the relationship between lipid accumulation and growth rate for microalgae. A high-lipid-accumulating microalgae *Chlorella vulgaris* NIES 227 was cultivated at pilot scale in order to characterize the joint evolution of its physiological status and photosynthetic productivity. Aiming at giving an accurate snapshot of the microalgae photosynthetic productivity, these measurements were performed by short assays of oxygen productivity under light-limited conditions.

## 2. Materials and Methods

### 2.1. Micro-Organisms and Culture Conditions

The microalga *Chlorella vulgaris* NIES 227 was selected for this study based on prior studies which demonstrated that this strain had the potential to accumulate lipid under nutrient limitation, and particularly nitrogen starvation [18]. The strain was cultivated in a 285 L photobioreactor Jumbo XL (Synoxis, Le Cellier, France). Agitation was ensured by compressed air bubbling, according to the manufacturer's directions. The culture temperature was controlled at 25 °C, and the pH was kept neutral (approx. 7.5), thanks to $CO_2$ bubbling. The microalgae were grown under natural sunlight inside a greenhouse in Saint-Paul-lez-Durance (France).

The microalgae were cultivated in semi-continuous batches, on 11 February 2021 and pure lab-grown cultures of the microalgae were suspended in the photobioreactor in nutritive medium (811 $mg \cdot L^{-1}$ of $NaNO_3$, 116.5 $mg \cdot L^{-1}$ of $KH_2PO_4$, 71.5 $mg \cdot L^{-1}$ of $MgSO_4$, $7H_2O$, 0.75 $mg \cdot L^{-1}$ of $CaCl_2$, $2H_2O$ and 0.5 $mL \cdot L^{-1}$ of Hutner solution [19] for trace elements). The microalgae cells were grown in batch cultures until harvesting. On the day of harvesting, approx. 95% of the cultivation volume was collected, and the remaining cells were resuspended in the same media as the inoculum, to start the following batch culture. Three harvests were performed for this study, implemented on 22 March 2021, 6 May 2021, and 15 June 2021.

The culture was regularly compensated for evaporation by adding ultra-pure water in the cultivation chamber until visually obtaining the adequate water level.

### 2.2. Laboratory Analysis

2.2.1. Sampling

Approx. 50 mL of cultivation was collected typically 2 to 3 times per week from a tap point off the side of the photobioreactor. The sample was collected before 9 a.m.

and immediately placed inside the refrigerator for later analysis. All analyses with fresh biomass described in the following sections were started within 6 h of collection.

### 2.2.2. Biomass Growth Monitoring

Biomass growth in the photobioreactor was monitored both in terms of dry-weight concentration and cell count in the microalgae culture. Dry-weight was determined according to the standard method 2540D [20], using GF/A or GF/C filters (Whatman™, Maidstone, UK). An 8% measurement error was considered for dry-weight measurement [21]. Biomass cell concentration was measured using a cell counter Multisizer 4 (Beckman Coulter™, Brea, CA, USA): briefly, approx. 20 µL of the culture (volume adjusted depending on the current cell concentration) was suspended in 20 mL of Isoton™ II Diluent (Beckman Coulter™, Brea, CA, USA). A study of duplicate measurements for cell counts (including measurements carried out outside of the scope of this study) evidenced a 15% measurement error on cell count. The measurement error retained for this study was therefore the most conservative value obtained from calculating the error obtained from a 15% error coefficient and the deviation measured from duplicates when available.

### 2.2.3. Biomass Metabolic Status

The biomass metabolic status was assessed, based on the content in proteins and energy-storage molecules (i.e., total carbohydrates and total lipids) of the biomass. The metabolic status of a microalga is discussed in terms of quota in proteins, quota in lipids, and quota in carbohydrates, i.e., the mass of proteins, lipids, or carbohydrates divided by (1) the corresponding mass of microalgae (henceforth defined as mass quota), or (2) the number of cells of microalgae (henceforth defined as cell quota).

### Proteins

The protein content was measured using the nitrogen content of the microalgae as a proxy, as commonly performed in the literature [22,23]. The nitrogen content of the biomass was determined from the centrifugation of a known volume of the culture ($4500\times g$ for 10 min in an Allegra X-15r centrifuge, Beckman Coulter™, Brea, CA, USA). The supernatant was discarded, and the pellet was resuspended in ultra-pure water and centrifuged again, to rinse the biomass. This process was repeated for a second rinse. The nitrogen content of the biomass was immediately determined, using a total organic carbon analyzer TOC-L (Shimadzu™, Kyoto, Japan), from the pellet resuspended in a known volume of ultra-pure water, to target a total nitrogen concentration in the final suspension within the range of the analyzer calibration. The protein quota of the algal biomass was computed assuming a nitrogen-to-protein ratio of 5.04, as reported for *Chlorella vulgaris* [24]. A 2.4% relative error was considered for the measurement of TN content of the biomass, based on Li et al. (2019) [25]. In order to compute the error on protein quota, a 1% relative error was considered for dilution factors, when applicable.

### Carbohydrates

On the day of sampling, known volumes of experimental culture were rinsed and centrifuged (as described for protein analysis) and freeze-dried (lyophilizer COSMOS 20K, Cryotec™, Saint-Gély-du-Fesc, France). The samples were hermetically sealed and stored in the freezer until analysis. Total carbohydrates were measured using the protocol developed by Dubois et al. (1956) [26]. Briefly, lyophilized biomass was digested in 2.5 M HCl (0.5 mL/g DW) at 100.5 °C for 3 h. The digestate was diluted in ultra-pure water ,to obtain measurable levels of total carbohydrates, and mixed with 2.5 mL of 95% $H_2SO_4$ and 500 µL of 5% (*w/v*) phenol solution. The total carbohydrate in the suspension obtained was determined using colorimetry, by comparing the light absorption of the solution at 483 nm with a calibration curve derived from a glucose standard solution (spectrophotometer Epoch 2, Biotek™, Winooski, VT, USA). The samples were typically measured in duplicate. Uncertainty as to the total carbohydrates in the biomass was based

on the standard deviation of replicates. When no replicate was available, a 20% relative error was assumed, based on the spread of the relative deviation observed from data with replicates.

Lipids

Total lipids were measured from freeze-dried pellets prepared according to the same protocol as used for the total carbohydrates analyses. Total lipids were measured by GC-FID (GC-2010 Pro AOC-20i/AOC-20s, Shimadzu™, Kyoto, Japan) following extraction and trans-methylation of the lipids. Briefly, the lyophilized biomass was mixed with 3 mL of a solution of 1.25 M hydrogen chloride in methanol (Reference 17935 Supelco, Sigma-Aldrich™, Saint Louis, MO, USA), 0.2 mL of a 3 mg·L$^{-1}$ solution of TAG C15:0 (reference T4257, Sigma-Aldrich™, Saint Louis, MO, USA) in hexane 95%, anhydrous for internal calibration of the measurement, and incubated at 85 °C for 1 h. A total of 3 mL of HPLC grade hexane was added; 1 mL of ultra-pure water was added to the sample, and the solution was briefly vortexed before centrifugation (1500 rpm, 5 min). The supernatant was introduced into vials for GC analysis. The total lipid quota was determined from the sum of the areas of all peaks detected, translated into a concentration based on the ratio of the area to the known quantity of C15:0 introduced using internal standard addition. Uncertainty regarding the lipid content in the biomass was based on the standard deviation of replicates. When no replicate was available, a 11% relative error was assumed, based on the spread of the relative deviation observed from data with replicates.

Functional Compartment

The non-null difference between the biomass dry-weight and the sum of the measured weights of proteins, total lipids, and total carbohydrates is henceforth referred to as the functional compartment [27].

2.2.4. Biomass Productivity Status

Biomass photosynthetic productivity was measured by oximetry using the OX1LP-6 (Qubit™, Kingston, Ontario, Canada) set-up, maintained at 25 °C by a recirculatory water bath (Haake SC100-A10, Thermo Fisher Scientific™, Waltham, MA, USA). The culture sample was diluted in fresh medium to obtain a test solution of optical density approx. 0.2–0.3 (spectrophotometer Epoch 2, Biotek™, Winooski, VT, USA). The test solution was degassed by bubbling N$_2$ gas for approx. 30 s, and immediately introduced into the OX1LP-6 cell. The cells were kept in the dark for 10 min, also enabling the temperature in the tested volume to equilibrate at 25 °C $\pm$ 1 °C. The light intensity was set at 100 μmol·m$^{-2}$·s$^{-1}$ in the constructor software. At time 0, data logging was started, to record dissolved-oxygen concentration variations during 10 min cycles, alternating 5 min in the dark and 5 min under light conditions. (The first 5 min of the data logging were obtained in the dark.) Three cycles were performed for each sample.

The biomass rates of oxygen production and of respiration were computed by linear regression of the oxygen-concentration variation against time, under light and in the dark, respectively. Biomass photosynthetic-oxygen-productivity was computed as the average of the three rates of oxygen production, each being compensated by the respiration rate recorded in the same cycle (i.e., during the dark phase and immediately prior to the illumination phase). Error regarding biomass photosynthetic-oxygen-productivity was computed as the standard deviation of the replicate obtained for each sample.

The biomass O$_2$ productivity was analyzed in light of the light intensity received by the biomass. An estimation of the photons effectively received in the experimental chamber of the set-up was performed as described in Supplementary Information S1. The light absorbed by microalgae during oximetric assays was evaluated as the number of photons of photosynthetic-active-radiation absorbed (generally expressed in μmol). Due to significant uncertainty of the value obtained, a conservative 20% relative uncertainty was used for this measurement.

### 2.3. Data Analysis

Data analysis, calculations, and the computer code (R 4.2.1. [28]) used for this study, can be found in an online repository [29]. The list of packages used for this study is given in Supplementary Information S2.

### 2.4. Complementary Measurements

The dataset provided in the online repository also includes the following complementary measurements:

- Sunlight intensity measured throughout the study duration. Sunlight intensity was measured in the greenhouse in the vicinity of the photobioreactor using the continuous data logging (data logger 2648A Hydra Series III, Fluke™, Everett, WA, USA) of a pyranometer CM21 (Kipp & Zonen™, Delft, The Netherlands). No data could be recovered for the period preceding the 9 March 2021, or the period ranging from 19 April 2021 to 27 April 2021).
- Total nitrogen in the dissolved phase measured from the filtration of the centrifugation supernatant of the culture (filters Puradisc 0.2 μm, Whatman™, Maidstone, UK). Total nitrogen in the dissolved phase was determined using the total organic carbon analyzer TOC-L (Shimadzu™, Kyoto, Japan).

## 3. Results

### 3.1. Photobioreactor Monitoring

Biomass-growth monitoring in terms of dry-weight and cell concentration (Figure 1) evidenced the following phases of microalgae growth: (1) an exponential growth phase, where microalgae cells divided exponentially, and (2) a limitation phase, where microalgae stopped dividing but the dry-weight concentration continued to increase, albeit at a diminished speed compared with the end of the first phase. Following harvesting and resuspension in fresh media, (3) a "relaxation" phase was observed for two to three days when the dry-weight would not increase as fast as during exponential growth, evidencing a lack of fitness of the resuspended microalgae cells. At the end of the exponential growth phase, a cell concentration of approx. 2 to $6 \times 10^{11}$ cells·$L^{-1}$ and a dry-weight concentration of approx. 3.0–3.7 g·$L^{-1}$ were reached. At the end of the limitation phase, a dry-weight as high as 6 g·$L^{-1}$ was achieved during the third batch, as the volumetric cell count did not vary significantly.

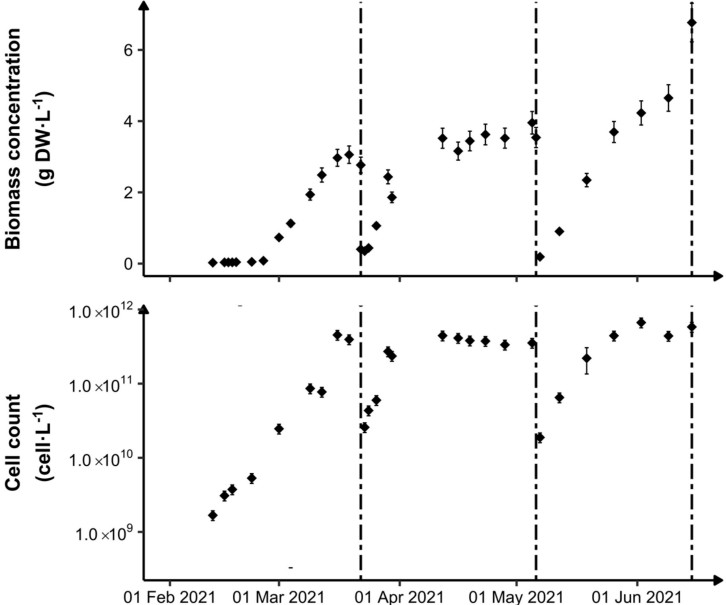

**Figure 1.** Evolution of biomass growth throughout the study. The dotted vertical line indicates the days the reactor was partially harvested.

### 3.2. Biomass Metabolic Status

As can be seen in Figure 2, the microalgae quota in protein during the exponential growth phase would initially remain high, followed by a gradual decrease (while still in exponential growth). In the three consecutive batches, the end of the exponential phase occurred when the protein-mass quota was down to the range (by batch chronological order) 24–28%, 18–31%, and 18–27%. The lipid-cell quota gradually increased when the protein-cell quota decreased. The end of the exponential growth phase occurred when the lipid-mass quota was in the range (by batch chronological order) 30–36%, 21–45%, and 25–31%. During the limitation phase, the protein-mass quota continued to drop, but appeared to stabilize at approx. $16 +/- 2\%$. The lipid-mass quota continued to increase, seemingly at a lower rate than during exponential growth. The cultivated *Chlorella vulgaris* NIES 227 reached up to $56.0 \pm 0.7\%$ mass percentage in extracted lipids during the second batch. In the relaxation phase (i.e., following resuspension in fresh media), the protein- and lipid-cell quotas initially recovered to levels close to the initial inoculum. The total carbohydrate-mass quota was somewhat constant throughout the cultivation period ($21.9 \pm 6.9\%$ of total dry-weight), and no particular pattern was identified for the variations measured. An increase in total carbohydrate content was noticed at the end of the third batch, potentially linked to the increase of dry-weight in the photobioreactor during the last sampling, despite a stable lipid-cell quota. The reasons for this sharp increase are unknown, but could be linked to experimental error.

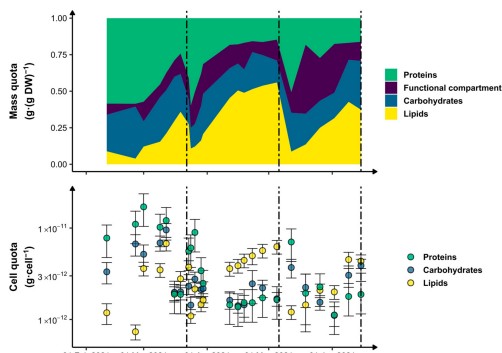

**Figure 2.** Storage molecule partition in *Chlorella vulgaris* NIES 227 biomass throughout the study. (It must be noted on the lower plot that the y-axis is represented in log scale).

Similar global culture dynamics were observed during the three batches used, yet some discrepancies in growth speed and physiological state of the culture occurred. It would be challenging to determine the exact causes of such differences: light intensity naturally varied during the different batches, and algae concentration at the start of the batch also varied, which was shown to affect the whole kinetics of the culture. Gradual acclimation of the culture to the experimental conditions may also have lessened the stress experienced by the microalgae cells throughout the study, as possibly indicated by the slower increase in lipid content during the third batch.

Overall, the photobioreactor yielded a total mass of lipids of 305, 565, and 758 g per batch, corresponding to lipid productivities of $2.8 \times 10^{-1}$, $4.4 \times 10^{-1}$, and $6.8 \times 10^{-1}$ g·m$^{-2}$·d$^{-1}$ during batch 1, 2, and 3, respectively, (assuming the photobioreactor has a 28 m$^2$ land footprint). If the culture had been harvested at the dates of maximum lipid productivity, productivities of $3.3 \times 10^{-1}$ g·m$^{-2}$·d$^{-1}$ (3 days prior to harvesting), $8.0 \times 10^{-1}$ g·m$^{-2}$·d$^{-1}$ (24 days prior to harvesting) could have been reached for the first two batches. (Batch 3 was harvested on the day of highest productivity recorded for this batch). Finally, the maximal lipid production rate based on consecutive measurements observed during this study was 1.00 g·m$^{-2}$·d$^{-1}$, during the penultimate week of cultivation before the harvesting of the last batch. Lipid productivity observed during this study was therefore significantly lower than the values reported in the introduction, which can be explained by the use of natural sunlight only, during the present study.

### 3.3. Biomass Photosynthetic Productivity

The parallel follow-up of the microalgae $O_2$ productivity and physiological state (Figure 3) demonstrated that the gradual loss of proteins and accumulation of lipids was co-occurring with a gradual decrease in the efficiency of photosynthesis. Hence, a 21-fold reduction in photosynthetic $O_2$ productivity (normalized for the calculated total light absorbed), ranging from $(1.27 \pm 0.26) \times 10^{-7}$ g $O_2 \cdot \mu mol^{-1}$ (lipid content of $53.8 \pm 0.2\%$, protein content of $16.3 \pm 0.4\%$, carbohydrate content of $21.0 \pm 2.8\%$), to $(2.75 \pm 0.62) \times 10^{-6}$ g $O_2 \cdot cell^{-1} \cdot \mu mol^{-1}$ (lipid content of $12.1 \pm 0.2\%$, protein content of $57.2 \pm 1.4\%$, carbohydrate content of $17.3 \pm 7.2\%$).

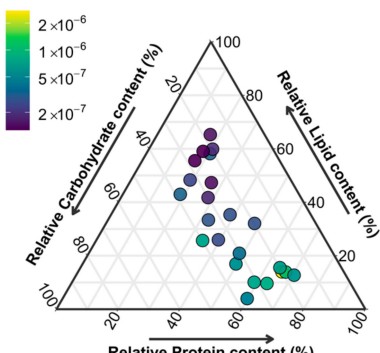

**Figure 3.** *Chlorella vulgaris* NIES 227 photosynthetic productivity under varying physiological statuses.

In particular, a strong relationship existed between the protein-cell quota and the photosynthetic response of the microalgae (Figure 4). A proposed predictive model was fitted from the data obtained through the linear regression between the log-transformed light-specific rate of oxygen production and the protein-cell quota (normalized to the minimal quota measured), leading to the relationship given in Equation (1).

$$P_{O_2}^* = P_{O_2}^{*,0} \cdot \left( \frac{q_p}{q_p^{min}} \right)^{\alpha} \tag{1}$$

where $P_{O_2}^*$ is the light-specific rate of oxygen production (g $O_2$ $\mu mol^{-1}$), $q_p$ is the protein quota in the microalgae cell (g Protein$\cdot cell^{-1}$), $q_p^{min}$ is the minimum protein-cell quota, evaluated at $1.12 \times 10^{-9}$ g Protein$\cdot cell^{-1}$. Values for $\alpha$ and $P_{O_2}^{*,0}$ were determined by linear regression of the experimental data, as described above ($R^2 = 0.632$, $p = 2.04 \times 10^{-6}$, N = 25): $P_{O_2}^{*,0}$ corresponds to the minimum light -pecific rate of oxygen productivity, determined as equal to $(1.59 \pm 0.27) \times 10^{-7}$ g $O_2 \cdot s^{-1} \cdot \mu mol^{-1}$, and $\alpha$ is a dimensionless parameter, evaluated as equal to $0.726 \pm 0.115$. Significant yet weaker correlations were found between the light-specific rate of oxygen production and the log-transformed lipid-cell quota ($R^2 = 0.223$, $p = 1.72 \times 10^{-2}$, N = 25, Figure S3), or the carbohydrate-cell quota ($R^2 = 0.291$, $p = 9.51 \times 10^{-3}$, N = 22, Figure S4).

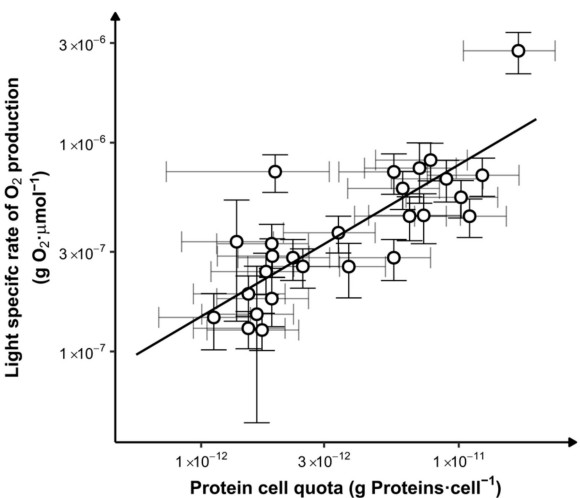

**Figure 4.** *Chlorella vulgaris* NIES 227 photosynthetic productivity according to the cell-protein quota (o). The continuous line represents the best fit of Equation (1) of the data collected ($R^2$ = 0.632, $p = 2.04 \times 10^{-6}$, N = 25).

## 4. Discussion

### 4.1. Light versus Nutrient Limitations

During the first batch of culture, the microalgal biomass first followed an exponential growth phase at constant protein-cell quota, but this incidence was not repeated in the second two batches. Because the initial concentration for the first batch was significantly lower than for the next two batches, we hypothesize that exponential growth at constant protein-cell quota was favored by growth conditions being neither light-limited nor nitrogen-limited. The microalga cells were therefore able to divide, while maintaining constant cell-specific nitrogen and carbon content, owing to a sufficient energy supply. As biomass cell density increased, the system entered a globally light-limited phase. Microalga cells continued to divide at a similar exponential rate, but the increase in dry-weight became linear. Although nitrogen was not depleted in the experimental culture (over 60 mg·L$^{-1}$ total nitrogen was measured in the supernatant at the onset of steady protein-content decrease, according to data provided in the online repository associated with this study [29]), the protein content started to decrease during this phase, and the lipid content to increase, highlighting a lack of nitrogen-assimilation capacity of the microalgae. Because the activity of nitrate reductase during nitrate assimilation for protein generation during photosynthesis has been previously shown to be light-dependent [30,31], light limitation in the experimental culture likely induced a globally reduced nitrogen-assimilation in the reactor, while maintaining a basal nitrogen uptake sufficient to enable cell division. The increase of lipid content, on the contrary, evidenced an excess of photosynthetically generated reducing power, diverted toward the accumulation of lipids following the mechanism described by Morales et al. (2021) [15]. As the culture further entered nutrient deprivation, microalgae cells stopped dividing and lipid content in the reactor increased. In the second and third batches, the ultimate value of lipid content in the microalgae was lower than the penultimate (in chronological order: reduction from 56.0 to 53.2% DW in 1 day, and from 42.7 to 37.3% DW in 6 days), indicating a possible limitation of the microalgae capacity for lipid accumulation. It is likely that as photosynthetic capacity decreased with progressing starvation, the carbon assimilation triggering the accumulation of lipids became overcompensated by carbon utilization through respiration processes. In all these phases, the total carbohydrate content in the microalgae remained somewhat stable.

### 4.2. Consequences for the Prediction of Lipid Accumulation

Microalgae convert photosynthetic energy into reducing power that will be involved in the conversion of $CO_2$ into carbohydrates through the Calvin cycle [32]. In parallel, the

reducing power is involved in the assimilation of the N-source (nitrate in the present study): broadly, nitrate is reduced to ammonium, and subsequently into organic N as glutamate [33]. We hypothesize from this study, the following dynamic from photosynthesis to model lipid accumulation:

I. The microalgae perform photosynthesis with an efficiency dependent on the current physiological state of the microalgae (i.e., partition between the cell content in carbohydrates, proteins, and lipids) and light availability, as evidenced by this study. This photosynthetic activity delivers a pool of electrons converted into carbohydrates [21].

II. The carbohydrates are stored and can subsequently be reduced to generate reducing power or be used as a carbon source for the following processes:

 a. Basic maintenance of the cell [34].

 b. Assimilation of nitrogen and protein formation: this study evidenced that the average light received by the cells likely modulates the quantity of nitrogen assimilated.

 c. Accumulation of lipids when excess carbon is absorbed in parallel with limited nitrogen assimilation.

The lipid accumulation by the microalgae *Chlorella vulgaris* NIES 227 may therefore be predicted by quantifying the total light energy absorbed by the microalgae cells in the experimental culture and determining the conversion factor of this light energy into reducing power, as performed in the present study through photo-respirometry. Following quantification of the nitrate assimilated by the microalgae (and the reducing power consumed in this process), the lipid generated can be calculated by the excess reducing power available.

### 4.3. Future Studies

4.3.1. Need for Short-Term Studies of the Evolution of Photosynthesis Performance

Modelling microalgae productivity during photo-autotrophic growth has received paramount attention from researchers, but few models have studied the physiological status of the microalgae in the prediction of biomass growth through photosynthesis. Among all the work cited in recent reviews of the literature [16,17], Geider et al. (1996) [35] and Kiefer et al. (1983) [36] modulated microalgae growth-rate from photosynthesis linearly, with the ratio (chlorophyll-a):(internal carbon storage) being an indicator of the microalgae cell fitness. Another common approach to integrate the modulation of photosynthetic growth through the physiological status of the cell has been the use of the Droop quota model [37], typically based on a down-regulation of microalgae growth rate by the cell quota in nitrogen [38–43]. All these approaches enable the prediction of the loss of gross productivity by the microalgae as it accumulates carbon (e.g., in the form of lipids), since it is equivalent to a reduction in the nitrogen-cell quota and chlorophyll content. These approaches and the Equation (1) developed in the study have an overall similar approach. However, the present study isolated, in order to evidence and quantify it, the loss of photosynthetic productivity of the microalgae, while all the studies cited above are based on "black-box" models, fitted over large amounts of parameters at once. The characterization of microalgae photosynthetic productivity as performed in the present study enables to estimate the pool of electrons a microalga is capable of generating for a given physiological state. The determination of full photosynthesis-irradiance (PI) curves to determine the variation of kinetic parameters according to cell physiological state is desirable for the later implementation in type II models, as described by Béchet et al. (2013) [16], although this was not the objective of the present study.

4.3.2. Nitrate Utilization Study

It was hypothesized from the present study that the nitrate utilization rate was likely to depend on the sunlight available under light-limiting conditions. Due to the mobilization of reducing power generated from photosynthesis during nitrate assimilation, the study of

the link between light availability and nitrate utilization is key for a better prediction of lipid accumulation in microalgae, particularly under light-limited growth.

### 4.3.3. Other Research Needs

As some of the electrons generated through photosynthesis are likely to be diverted toward metabolic pathways other than carbohydrate formation (e.g., acidification of the lumen [44]), a study to quantify such losses is desirable. A specific study of the rate of conversion of carbohydrates into lipids for energy storage by the microalgae, investigating the factors associated to its variations, is also needed. Refining studies could explore the characteristics of cell maintenance (i.e., the utilization of internal carbohydrates in the absence of energy supply [34]) and decay (death rate). Finally, any factors known to induce cell stress (e.g., phytohormones, metallic ions) could be investigated, to quantify both the impact on the photosynthetic activity of the microalgae and the impact on the carbon allocation pathways toward the different storage molecules.

### 5. Conclusions

PI-curves have been successfully used to predict microalgae productivity cultivated in nutrient-replete conditions [45]. As this study evidenced and quantified the gradual loss of photosynthetic activity from the microalgae *Chlorella vulgaris* NIES 227 while accumulating lipid during growth limitation, it is expected that calibrating PI-curve changes with varying microalgae metabolic statuses could enable to the precise prediction of lipid yields for a photobioreactor in changing culture conditions, representative of the conditions needed to induce lipid accumulation.

**Supplementary Materials:** The following supporting information can be downloaded at: https://www.mdpi.com/article/10.3390/fermentation8110614/s1: Supplementary Information S1: Estimation of the photon flux utilized by microalgae during oximetric experiments; Table S1: Light absorbance for the wavelengths 880 nm and 683 nm of *Chlorella vulgaris* NIES 227 cultures serially diluted used for the calibration of light field determination in the experimental chamber of OX1LP-6 set-up. Figure S1: Graphical representation of the experimental chamber of OX1LP-6 set-up and the sampling points for light intensity Table S2: Light intensity as measured in each sampling point in the experimental chamber according to the dilution of the microalgal culture tested. Figure S2: Light intensity in light sampling points 1, 2, and 3 according to the light absorbance at 880 nm of the microalgal solution. Table S3: Summary of fitting results and performance for each light sampling point. Supplementary Information S2: R™ packages specifically used during the study; Supplementary Information S3: Correlation between $O_2$ productivity, and lipid and carbohydrate cell quotas. Figure S3. Chlorella vulgaris NIES 227 photosynthetic productivity according to the cell lipid quota. Figure S4. Chlorella vulgaris NIES 227 photosynthetic productivity according to the cell carbohydrate quota.

**Author Contributions:** Conceptualization: P.C. and F.D.; methodology, P.C., F.D., A.D.-L. and A.R.-R.; software, P.C.; formal analysis, P.C.; investigation, P.C.; resources, F.D., A.D.-L. and J.-F.S.; data curation, P.C.; writing—original draft preparation, P.C.; writing—review and editing, P.C., A.R.-R. and F.D.; visualization, P.C.; supervision, F.D.; project administration, F.D. and J.-F.S.; funding acquisition, F.D. and J.-F.S. All authors have read and agreed to the published version of the manuscript.

**Funding:** This project has received internal funding from the Carbon Circular Economy program of Comissariat à l'Energie Atomique et aux Energies Alternatives, Paris, France.

**Institutional Review Board Statement:** Not applicable.

**Informed Consent Statement:** Not applicable.

**Data Availability Statement:** The data and code associated with the results presented in this study can be found in the online repository https://doi.org/10.6084/m9.figshare.c.6229317.v1 (accessed on 4 January 2022) [29].

**Acknowledgments:** The authors wish to thank Gatien Fleury (CEA) for his work in the development of the lipid analysis method used during this study, and his input into the revision of the manuscript.

**Conflicts of Interest:** The authors declare no conflict of interest.

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
