# Peer review of "Photosynthetic Carbon Uptake Correlates with Cell Protein Content during Lipid Accumulation in the Microalga Chlorella vulgaris NIES 227"

_fermentation, doi:10.3390/fermentation8110614_

Round 1

Reviewer 1 Report

The manuscript entitled " Photosynthetic carbon uptake correlates with cell protein content during lipid accumulation in the microalgae Chlorella vulgaris NIES 227". The findings are interesting and offer good perspectives as to predict protein or other components of cells by oxygen productivity of microalgae. However, some unclear parts should be clarified. Additionally, the writings, especially grammar mistakes and incorrect tense, need to be revised globally over the manuscript. Here are some specific comments.

In the title, correct “microalgae” to “microalga”.

Some grammar mistakes or Inappropriate format: Line 30; Line 75; Line84-85; line 284.

Line 19: “±” should be consistent with others.

Line 20-21: the manuscript only gave the linear model relating oxygen productivity and protein content. I think it’s hard to give the conclusion of “Quantified the link between biomass productivity dynamics and lipid accumulation”.

Line 80-81: I suggest that illumination intensity during culture should be given. It’s an important parameter for biomass and lipid accumulation of microalgae.

Line 198 to 206: The every cultivation process of 3 batches was divided into exponential growth phase, starvation phase and relaxation phase. When did the three phases in 3 different batches start? How to confirm the timing of starvation phase?

Line 286: The protein started to decrease along with lipid increasing when the nitrogen concentration was over 60 mg/L in the medium. However, the authors concluded that lipids accumulated during nitrogen starvation. It’s widely known that nitrogen starvation changed physiological state of microalgae including lipid and protein. Therefore, nitrogen concentration during cultivation should be given in the manuscript.

Line 231-232: Did you repeated the experiment?

Line 370: I think this part is more of a prediction than a conclusion.

Reviewer 2 Report

General comments:

This study addresses how different physiological statuses of a photosynthetic microalga are associated with each other and particularly emphasizes that photosynthetic activities could be used as a key parameter when modeling algal productivities. Based on the results, protein cell quota (g proteins/cell) should be primarily considered because of its strong positive correlation with photosynthetic activity (as measured in oxygen generation here), the decrease of which seems to be closely associated with the onset of lipid accumulation. Overall, this study is well-carried out and is clearly organized. Minor suggestions for the authors are listed below:

1.    Line 115: “respectively lipids or carbohydrates” reads strange. The authors provided cellular quota for each major biochemical component in Fig 2 – Please revise this for more clarity.

2.    Line 125: Does TOC analyzer give similar results as a typical elemental analyzer (i.e., Dumas method-based analyzer)?

3.    Line 165: Functional compartment seems to better defined. Is the majority of functional compartment ash? Or are there any recalcitrant components (e.g., non-hydrolyzable polysaccharides) existing at a substantial quantity?

4.    Line 188: thus should be revised.

5.    Lines 231-232: Would it be also possible that NIES 227 acclimated (or even adapted) to the conditions which a series of semi-continuous cultivation was performed? This could be partially associated with the lower FAME yield in the third batch, considering lipid accumulation as a response against stress conditions (in other words, the last batch experienced a lesser degree of cellular stress). Of course, this is just a speculation, but worth to be discussed somewhere in the manuscript.

6.    Figure 4: A second figure that has lipid cell quota (or 1/[lipid cell quota]) as X-axis could be added here to see whether there is more strong (or weaker) correlation between two axes.

7.    Was light intensity measured during the cultivation period? Combining the actual light intensity experienced during the cellular cultivation into the model could improve the model developed in this study.   

8.    Line 303: One thing that should be considered is that carbon assimilation and utilization may not be under a direct competitive relation. Similarly, there could be a dynamic change in the accumulated lipids (even though it is true that the overall rate is biased towards lipid accumulation under N-depletion). A piece that should be elucidated further is cellular stress level and how it influences the onset of lipid accumulation along with other stress responses.

Round 2

Reviewer 1 Report

Line 20-21: the manuscript only gave the linear model relating oxygen productivity and protein content. I think it’s hard to give the conclusion of “Quantified the link between biomass productivity dynamics and lipid accumulation”.

The whole sentence was replaced by (L21-23) The decrease in photosynthetic Oproductivity was evidenced to correlate to the cell internal protein content (R= 0.632, p = 2.04·10-6, N = 25)”.

You did give the correlation between photosynthetic Oproductivity and cell protein, but there is no data with Oproductivity and lipid content, which you displayed in Line 21-23.

Reviewer 2 Report

The authors improved the manuscript substantially. No further comments here. 
